# Understanding the perceptions of parents and preschool principals on the determinants of weight management among Iranian preschoolers: A directed qualitative content analysis

Najmeh Hamzavi Zarghani[1], Fazlollah Ghofranipour[1]*, Eesa Mohammadi[2], Greet Cardon[3]

**1** Department of Health Education and Health Promotion, Faculty of Medical Science, Tarbiat Modares University, Tehran, Iran, **2** Department of Nursing, Faculty of Medical Science, Tarbiat Modares University, Tehran, Iran, **3** Department of Movement and Sports Sciences, Ghent University, Ghent, Belgium

* ghofranf@modares.ac.ir

**Editor:** Éadaoin Butler, Trinity College Dublin, IRELAND

**Data Availability Statement:** All relevant data are within the paper and its Supporting Information files.

## Abstract

The current study aimed to understand the perceptions and experiences of Iranian parents and principals of preschool children on weight management based on the PRECEDE-PROCEED Model (PPM), a comprehensive structure for assessing health needs for designing, implementing, and evaluating health promotion, and other public health programs. PRECEDE provides a structure for planning a targeted and focused public health program, and PROCEED provides a structure for implementing and evaluating the program. Data were gathered from 17 preschoolers' parents and two principals using semi-structured interviews in the preschool setting in Tehran, the capital of Iran, in 2019. Data were analyzed manually through directed content analysis based on constructs in phases two and three of the PPM, simultaneously with data collection. This study identified genetic, behavioral (e.g., food preferences, physical activity, sedentary behaviors, the effect of parents', peers', principals' and teachers' behavior and also influence of grandparents' and neighbors' behaviors) and environmental (e.g., home, grandparents' home and preschool) factors from the epidemiological construct. Also, predisposing (e.g., child's attitude, parent's and principals' attitude, as well as parents' knowledge and parents' and principals' beliefs), enabling (e.g., parental skills and skills of the principals and teachers, rules and laws in the preschools, and availability), and reinforcing (e.g., family support and influences, teachers' encouragement and influences, and peers' influences) factors were identified from the educational and ecological construct. Additionally, "quality of child-parent relationship" was determined as a new factor affecting preschoolers' weight management promotion; however, it was not in the PPM. In the study, parents' and principals' experiences regarding preschoolers' weight management promotion confirmed the genetic, behavioral, environmental, predisposing, enabling and reinforcing factors of the PPM. "Quality of child-parent relationship" factor may be related to the culture and family relationship type of Iranian people, which is suggested to be investigated in future studies.

**Funding:** The authors received no specific funding for this work.

**Competing interests:** The authors have declared that no competing interests exist.

# Background

The high prevalence of overweight and obesity in children has become a public health problem in developing and developed countries [1]. The prevalence of overweight (including obesity) among preschoolers was 32.0% and 10.0%- 20.6% in the United States and Europe, respectively [2, 3]. A systematic review and meta-analysis conducted on Iranian children and adolescents showed an increasing trend in weight gain [4]. Additionally, the prevalence of overweight and obesity was relatively high in both male and female preschoolers in Tehran, Iran [5].

The increased risk of physical disorders, such as cardiovascular diseases, cancer, liver steatosis and psychological disorders, like low self-esteem, body image concerns, depression, and weak socialization, can be linked to overweight during childhood [6–9]. Therefore, early childhood has been recognized as a crucial period to determine the risk factors of obesity and establish a healthy lifestyle to prevent obesity and other chronic diseases and their complications [5, 10].

In many studies, health care professionals and parents have reported several barriers and facilitators of weight management programs. Some parents may not foster healthy weight management for their children due to 1) low level of knowledge about the complications of overweight/obesity, 2) underestimation of the weight status of their children and 3) fear of the stigma associated with obesity [11–13]. On the other hand, some studies have reported facilitators to improve weight management, such as better parenting skills regarding healthy behavior, being a role model for children, parents' concerns about children's health, particularly their psychological status, e.g., higher self-esteem and self-confidence [13, 14].

With a population of 83 million people, Iran, that is located in Western Asia, is the world's 18th most populous country. Studies on the weight status of Iranian preschoolers are limited to descriptive reports of the prevalence of overweight/ obesity or related risk factors. In Tehran, the prevalence of overweight and obesity among children aged 3–6 years was 10.3% and 4.5% in girls and 9.8% and 4.7% in boys, respectively [15]. Other studies have revealed overweight/ obesity as a health problem among Iranian preschoolers and reported their association with some variables, such as low physical fitness and high levels of screen time in children, and parental obesity [5, 16]. Parents play an important role in shaping their children's behaviors [17]. Therefore, understanding parents' perceptions of their children's weight status and its determinants are important to develop strategies and programs for weight management [18]; however, these perceptions have not yet been investigated in Iranian parents.

The preschool setting can also play an important role in children's weight management [19]. For example, in preschools of Tehran, the principals arrange and manage the diet and physical activity programs. Therefore, along with the perceptions of Iranian parents, assessing the perceptions of Iranian preschool principals on weight management of preschool children is of great importance that should be considered for the development of weight management strategies and programs.

The PRECEDE-PROCEED Model (PPM) commonly provides practical guidance for health education and health promotion [20]. The current study mainly focused on the second (epidemiological assessment) and third (educational and ecological assessment) phases of the PPM to better understand the health problem and potential modifiable strategies in the Iranian family and preschool context. Other stages are also important; however, are beyond the scope of the current study due to time and tools restrictions.

In the second phase, *epidemiological assessment*, health problems and their causative factors, including genetic, behavioral, and environmental factors, are recognized [20]. The behavioral determinants include three levels: proximal- direct actions influencing one's own health; actions influencing the health of others- and distal actions influencing organizational or policy

environment [21]. Environmental factors, such as physical and social determinants, are factors outside the person that can be modified to support behavior, quality of life, or health [21]. The third phase, *educational and ecological assessment*, contains three predisposing, enabling, and reinforcing factors [20]. Predisposing factors include awareness, knowledge, attitude, beliefs, values, and perceptions of individuals that facilitate or inhibit motivation for change [20]. These factors also contain early childhood experiences that establish values, attitudes, and perceptions in the child's first place of residence [21]. Enabling factors, such as accessibility, availability, community resources, laws, and skills, are identified as readiness for behavioral and environmental change [20, 21]. Reinforcing factors provide rewards or feedback for adopting and maintaining a particular behavior (such as reinforcement by family members, teachers, health care staff, peers, and community leaders) [20, 21].

The purpose of the present study was to understand the perceptions and experiences of parents and principals of Iranian preschoolers on weight management and explore genetic, behavioral, environmental, and predisposing, reinforcing, and enabling factors of weight management-related behaviors on the PPM.

## Methods

### Ethics approval and the consent to participate

The current study was approved by the Ethical Committee Board of Faculty of Medical Sciences of the Tarbiat Modares University (Approval code: IR.MODARES.REC.1397.034). In order to comply with ethical standards, the researcher explained the objective and methodology of the study to the participants and written informed consent received. The interviews were transcribed without reporting identifying information and names and participants were reassured about confidentiality and anonymity.

### Study design

This qualitative study using a directed content analysis approach [22] based on the PPM was done on preschoolers aged 3–5 years in Tehran in 2019. We used individual semi-structured interviews with open-ended questions, and assessed play equipment and documents, such as the diet and physical activity plans in preschools. This triangulation method increases the credibility and conformability of the study and the understanding of various aspects of weight management promotion. Furthermore, the research team explored genetic, behavioral, environmental, educational, and ecological factors, including predisposing, enabling and reinforcing of weight management.

### Study setting and participants

Purposive sampling was used to choose participants with the maximum variation based on sex, age, educational and occupational status (see Tables 1 and 2). Preschools in six areas of Tehran divided by the Municipality of Tehran were selected. Although the preschools were supervised by the Welfare Organization of Tehran, parents and preschool principals provided equipment and food preparation costs. Based on the literature, parents, especially mothers, are the most important agents in children's weight management [1, 7, 18] Also, in Iran, mothers are more involved in taking care of their children and their eating habits and physical activity than fathers. Therefore, the research team decided to prioritize interviewing mothers. The inclusion criterion was being a principal or parent of preschoolers aged 3–5 years. The study sample size included 19 individuals; mothers (15), fathers (2), and preschool principals (2).

**Table 1. Demographics characteristics of parents.**

| Variables | Number | Percent |
|---|---|---|
| **Sex of participants** | | |
| Female | 15 | 88.3 |
| Male | 2 | 11.7 |
| **Sex of child** | | |
| Female | 11 | 64.7 |
| Male | 6 | 35.3 |
| **Educational status of Participants** | | |
| Bachelor and lower | 13 | 76.5 |
| Postgraduate education | 4 | 23.5 |
| **Occupational status** | | |
| Housewife | 4 | 23.5 |
| Employed | 13 | 76.5 |

Two mothers said that they did not have enough time to interview; therefore, they refused to participate in the study.

## Data collection

Data were collected through a triangulation method, including individual semi-structured in-depth interviews, assessment of preschool playground equipment, as well as the documents, such as existing diet and physical activity plans in preschools. The Interview questions were based on the epidemiological (including genetic, behavior and environment) and educational and ecological (including predisposing, enabling, and reinforcing factors) phases of the PPM. Table 3 presents the interview questions answered by the parents and principals. The interviews were executed by the trained researcher in the preschool setting and lasted 15 to 35 minutes. After the participants answered, the researcher utilized probe questions to explore participants' experiences of the concepts by asking questions, such as "What do you mean?, Please explain more about . . .?, Could you give me an example to understand what you mean?". After transcription and review of some interviews, to clarify ambiguities, the researcher called the participants and asked them for more details. Before each interview, a written informed consent form was signed by all participants, in which they allowed the recording of the interviews.

## Data analysis

Audio recording of the interviews was done by the researcher, and then, they were transcribed verbatim. The interviews were reviewed and manually coded based on the determined constructs of the PPM by two authors. In line with the study's goals, the researchers utilized the approach developed by Hsieh and Shannon for directed content analysis [22]. After reading each interview several times and understanding it deeply, the texts, which seemed related to weight management at the first impression, were coded with the predetermined codes. The researcher designed summary sheets based on the structures of the model and embedded the

**Table 2. Demographics characteristics and work experience of two principals of preschools.**

| Participants | Age | Sex | Educational level | Work experience |
|---|---|---|---|---|
| **Principal 1** | 59 | female | Bachelor | 35 |
| **Principal 2** | 46 | female | Bachelor | 15 |

**Table 3. Interview questions.**

| Constructs of PPM | Interview Questions | Audiences |
|---|---|---|
| **Behavioral factors** | 1. Please explain your child's nutrition status. | parents |
| | 2. Please explain your child/ children's physical activity on weekdays and weekends. | parents/principals |
| | 3. How do you manage your child/ children's weight? | parents/principals |
| **Predisposing factors** | 4. How do you assess your children's weight? (Based on the practitioner's assessment or your own assessment) | parents |
| | 5. Why do you manage your child/ children's weight? | parents/principals |
| | 6. What is your belief regarding children's weight status aged 3–5 years? | parents/principals |
| **Enabling factors** | 7. Please tell me about the rules in your home to manage your child's weight. | parents |
| | 8. Please tell me about the preschool's dietary and physical activity plans. | parents/principals |
| | 9. What are the rules in the preschool to improve children's weight management? | principals |
| | 10. What problems do you encounter to manage your child's weight? | parents |
| **Reinforcement factors** | 11. Please tell me how you influence your children's dietary intake and physical activity. | parents |
| | 12. Please tell me how teachers and peers influence your child/ children's eating habits and physical activity. | parents/principals |

codes in the summary sheets to determine which categories were most approved by the interviewees. Data unrelated to these categories were given a new code and then were newly categorized. The researcher utilized member checking to ensure that the analyses can indicate the participants' experiences and perceptions. The PPM was used as an external check for the dependability and conformability of the data. An instance of coding and putting in categories and subcategories is presented in Table 4.

## Results

The mean age of parents was 35.1 (range: 28–42) years, and the principals' mean age was 52.5 (range: 46–59) years. The categories of genetic, behavioral, environmental, predisposing, enabling and reinforcing factors contained 2, 67, 24, 83, 33 and 17 codes, respectively. Fifteen codes were not placed under any category or subcategory in the PPM; however, participants considered some related to children's weight management. Therefore, the research team considered a new category for these codes. This new category was named "Quality of child-parent relationship" and contained a subcategory "the effects of maternal mood status and quality of parental relationship on child's weight-related behaviors". Data saturation was considered when the last couple of interviewees did not add new perceptions and sufficient data had been obtained regarding the object [23]. After 16 interviews, the codes were repeated, no new codes were found, the data were considered saturated, and data collection was stopped. A summary of the content analysis results related to genetic, behavioral, environmental, predisposing, reinforcing, and enabling factors in the PPM is shown in Fig 1.

### 1. Genetic factors

Genetic factors play a role in obesity and also lack of obesity. Although genetic predisposition affects obesity, an obesogenic environment increases the risk of obesity [24]. Few mothers and one of the principals mentioned "history of appetite and slimming in the family" as a determinant genetic factor for weight management (see Table 4).

### 2. Behavioral factors

Based on direct actions affecting one's own health in this model, data analysis revealed "food preferences, not eating some foods due to allergy or stomach reflux, and physical activity and sedentary behaviors" as behaviors performed by children.

**Table 4. The coding matrix table based on the PPM.**

| Category | Subcategory | Code | Meaning unit |
|---|---|---|---|
| Genetic factor | - | History of appetite and slimming in the family | "Maybe genetic factors affect it, because my sisters and I were the same as my daughter when we were a child" |
| Behavioral factor | Proximal-actions influencing one's own health | Food preferences | "He always eats cake for a snack . . ." |
| | | Sedentary-based behaviors | "Every day she watches TV from 1:00 PM to 4:30 PM. . . And also she makes handicrafts, and she draws"" |
| | proximal- actions affecting the health of others | The effect of parents' behaviors | "She doesn't have onion like her father . . ." |
| | | The effect of principals' and teachers' behaviors | "My daughter said: I was reluctant to eat my snack (carrot) but aunt (her teacher) said to eat it, and I said to myself it was a pity and I ate it" |
| Environmental factor | Physical environment | Eating unhealthy snacks at home | "Yes, she eats unhealthy snacks at home, completely" |
| | | Eating food meals in the preschool | "Yeah, breakfast, snacks and lunch. After lunch, again, I prepare something for her; she says I am hungry" |
| Predisposing factor | Attitude | Parents' satisfaction with their children's activity | "Her physical activity is tremendously high. Every moment she comes home, she plays a lot until she goes to bed" |
| | | Mothers' dissatisfaction with children's food consumption | "Due to the high metabolism in my child's body, I expect to eat a lot more, but not" |
| | Beliefs | Belief in having normal weight | "I always say a child has to be healthy, not overweight. . ." |
| | | Eating food in a group, very well | "If children don't eat at home due to coercion, they will have very well in the preschool. The most reason is that they eat meals together" |
| Enabling factor | Law and rules | Playing in the yard of preschool | "If it is not snowy or rainy, they wear their cloths and play in the yard. |
| | | Preparing a weekly food plan | "I write a weekly food plan and it is approved by a nutrition consultant. She has a seal" |
| | Skills | Paying attention to preschoolers' health | ". . . cotlet (a mixture of meat, potatoes, and eggs), I don't really agree that a child eats it because of frying" |
| | | Showing children healthy eating behaviors | "There is some food that her mother and I do not like, but we do not show that in the appearance . . ." |
| Reinforcing factor | Feedback, encouragement and influencing of others | Teachers' influencing on children's eating and physical activity behaviors | "Teachers are also effective because my daughter constantly emphasizes what food items are useful and what food is not beneficial . . ." |
| | | Influencing peers' eating behavior | "For example, if one day one of the children doesn't eat soup, when she comes home she says won't eat that" |

**2.1. Children's actions.** *2.1.1. Eating behavior.* Mothers reported that many children preferred sugar-sweetened beverages, cake, and unhealthy snacks, and sometimes they liked to go out to eat pizza. Few children had consumed fruits and homemade juice as a snack. *"He always eats cake for a snack. If there is no cake in his package, he is stubborn and says I don't go to the preschool at all . . ."* (mother of a 5-year-old boy). However, one mother said *"every day, my son eats fruit, completely"* (mother of a 5-year-old and 1 month boy). Some mothers stated that their children had a restriction on eating some kinds of food due to asthma or reflux. "He has asthma, and I ask the preschool principal not to give my son certain food" (mother of a 5-year-old boy).

*2.1.2. Physical activity and sedentary behaviors.* Most mothers mentioned that their children had a lot of physical activity and played with their bicycles or scooters and playground equipment in parks. *". . . Some children are very slow; however, my daughter is very active"* (mother of a girl aged three years and eight months). Many mothers also stated that their children often performed sedentary behaviors, such as brain games and using the screen at home. *"Every day she watches TV from 1:00 PM to 4:30 PM. . . And also she makes handicrafts, and she draws"* (mother of a 5-year-old girl). A preschool principal said that if weather was good, children would play in the preschool yard under the teacher's supervision. *"Some days we will have a playing program in the yard if the weather circumstances are favorable and it is not polluted. Children go to the yard and run. There is some playground equipment . . ."* (principal, code 12).

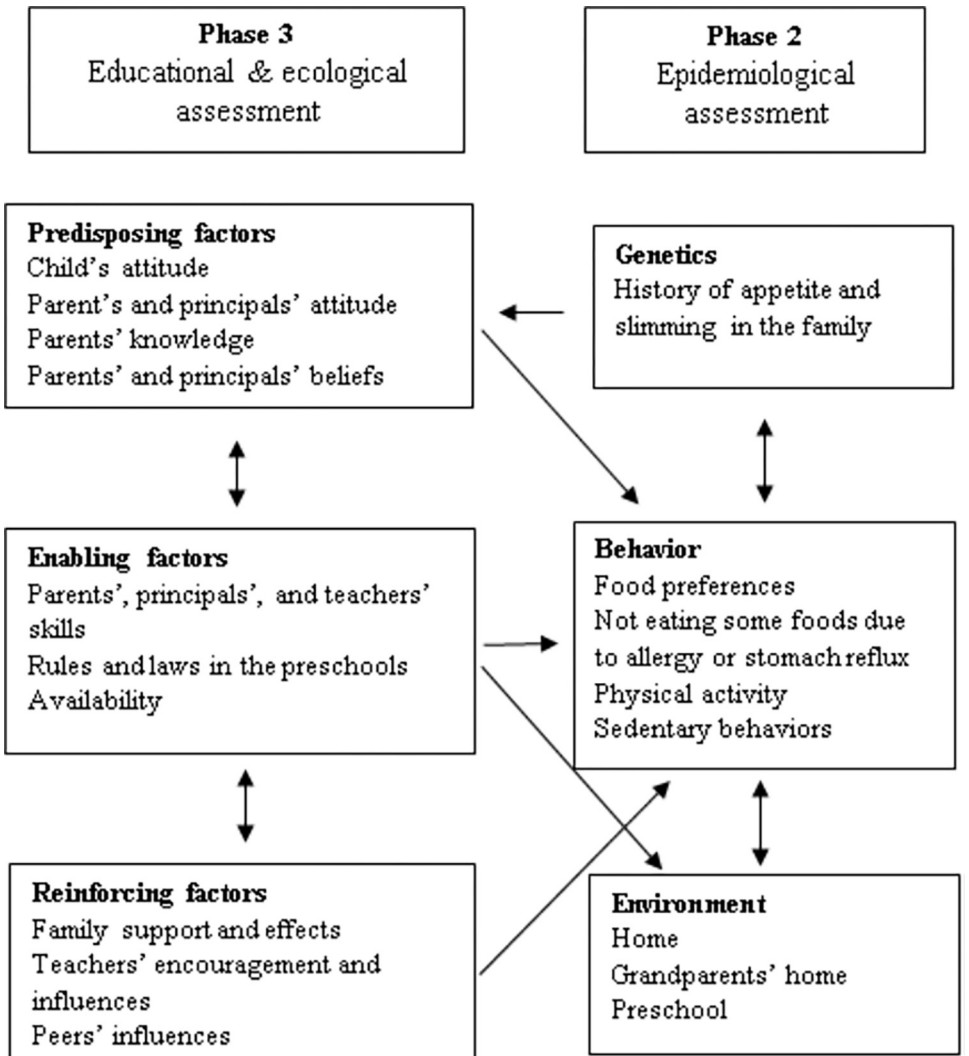

**Fig 1. PRECEDE-PROCEED model to understand perceptions and experiences of Iranian parents and principals of preschool children on weight management.**

**2.2 Actions of others.** Another category of behavioral factors included actions affecting the health of others: "the effect of the behavior of parents and peers; the effect of the behavior of principals and teachers in the preschool, and the influence of grandparents' and neighbors' behaviors."

*2.2.1. Actions of parents.* More mothers admitted that they forced-feed their children, and even fathers endorsed and complained about it. *"Her mother, yes, but I say whenever she is hungry, she comes and says it to us . . ."* (father of a girl aged four years and two months). Although few mothers reported no use of coercion, or some of them reported using it before, it was not currently used after finding its disadvantageous. *"Even I wanted to force-feed them, so that they would get nausea, they even vomited what they had eaten. I preferred not to give them anything forced"* (mother of a girl aged 5 years and 6 months).

Parents also reported they were responsible for taking their children to the preschool by car or on foot. Families whose homes were close to preschools and housewives mothers usually took their children to the preschool on foot. *"Myself. Because our route is close to the preschool I myself take him in the mornings, and I take him home at noon"* (mother of a 4-year-old boy).

Mothers said in cases where the father was in charge of taking the child to the preschool, or when the distance was long between home and the preschool, parents took their children by car. *"We get in a taxi. From here to home we can't walk because it is far away"* (mother of a five years and one month boy). Some mothers reported that they took their children to the park, and also a few children's fathers took them to the club or pool. *"I take my daughter to the park once a week every summer"* (mother of a four years and six months old girl). Some mothers explained that they gave children a cell phone or tablet to entertain them and to lessen their mischief, and also mothers could handle daily tasks. *". . . His father and I have a homebuilding program on our phones, and if he bothers us, we will give him our cell phone for ten minutes or a quarter"* (mother of a 5-year-old boy). In addition, some parents reported doing sedentary activities, such as painting, crafts and playing with Lego bricks with their children, because they were living in apartments and had some limitations for noise and also mothers were not eager to play active games, such as wrestling and jumping. *"He knows I'm not a person to play active games, he says mom, let's play a brain game. He says I know you like sitting game, which game"* (mother of a boy aged five years and six months).

*2.2.2. Peers' actions.* Most mothers and principals reported that dietary behaviors of peers affected children; however, some mothers revealed that peers' dietary behaviors did not affect their children's behaviors. Peers affected most children when they did not like a kind of food and they did not eat it; thus, the child tried doing that behavior. *"For example, if a child doesn't have soup, she comes home and says I will not eat soup, but she doesn't insist on it"* (mother of a girl aged five years and 10 months). Another mother said *"No, for example, my family's children don't influence my son at all"* (mother of a 5-year-old boy). A positive effect of peers regarding dietary behaviors was to improve nutrition due to eating as a group of children in the preschool. *"She is getting better now. Until she did not come to preschool, she didn't eat food at all. . . But since she comes to the preschool she sees kids, and she gets better"* (mother of a girl aged three years and seven months).

*2.2.3. Actions of principals and teachers.* Mothers and one of the principals mentioned the influences of teachers on children's behavior, especially dietary behaviors. *"My daughter said: I was reluctant to eat my snack (carrot) but aunt (her teacher) said to eat it, and I said to myself it was a pity, and I ate it"* (mother of a girl aged five years and 11 months). The principals reported caring about the children's health and nutrition and tried to use high-quality foods to prepare children's meals. *"We have chefs. Cooking is performed according to the approved food program here. I myself buy foodstuffs from a safe store. . . In addition, we don't use bulk food because they are contaminated"* (principal, code 12).

*2.2.4. Actions of grandparents and neighbors.* The interferences of grandparents in the feeding of children were mentioned by some parents. *"When my maternity leave was over, and I went to work, my family pressured on me more, and they said, you were going to work, and Paria became foodless, look, the child became slim. They pressured me, and I pressured my daughter . . ."* (mother of a girl aged three years and eight months). Also, mothers said that one of the reasons for restricting children from playing and doing physical activities is some limitations of living in an apartment. They said we have to play sedentary games with our children instead of running because our neighbors are bothered by the noise of children. *"Thinking games, sitting activities. . . She plays running games to the extent that the apartment is allowed to play"* (mother of a girl aged three years and seven months).

## 3. Environmental factors

Based on data analysis, the researchers identified three environments, including "home, grandparents' home, and preschool" in the subcategory of the physical environment.

**3.1. Physical environment.** *3.1.1. Home.* Spending time on screen-based behaviors, including television, a mobile phone, or a tablet, was common among children at home. A few mothers reported spending a lot of time watching TV with their children. *". . . If he stays awake 10 hours a day, he watches TV for six hours"* (mother of a boy aged three years and six months). However, a few mothers also reported that children had a lot of activities at home. *"At home, she goes over to the desk or table and she jumps on the furniture"* (mother of a girl aged five years and 10 months). At home, children were reported eating some kinds of food, such as unhealthy snacks, *"She eats unhealthy snacks completely at home . . ."* (mother of a girl aged five years and six months).

*3.1.2. Grandparent's home.* Some parents reported their children playing with their peers at grandparents' homes, other children in their neighborhood, or with relatives' children. *". . . There are a lot of playmates at my mom's building, and she plays with them at the weekend"* (father of a girl aged four years and two months).

*3.1.3. Preschool.* Children spend most of the days in a week and most of the hours in a day in the preschool and were eating breakfast, lunch, and snacks there. *"Yeah, breakfast, snacks and lunch. After lunch, again, I prepare something for her; she says I am hungry"* (mother of a girl aged five years and 11 months). Mothers and principals pointed out that children prefer eating their meals in a group and with cravings in the preschool. *"Some of them don't have meals well. . . . but because they are in a group, they are encouraged to eat. When they see all the children are eating a kind of food it will be an encouragement for them to eat"* (principal, code 12). In the preschool, the principal stated that children were allowed to bring Pofila (Popcorn) as the only snack, and also they had fruit, nuts, packaged milk, and biscuits for the snack. *"Snacks, they can bring Pofila only"* (principal, code 12).

# 4. Educational and ecological factors

**4.1. Predisposing factors.** Among the predisposing factors, according to data analysis, "the child's attitude, parent's and principals' attitude, as well as parents' knowledge and parents' and principals' beliefs" were explored.

*4.1.1 Attitude. 4.1.1.1. Children's attitude.* The results showed that some children preferred playing and doing physical activities to using mobile phones or playing with dolls or watching TV, and also, most of them tended to have a playmate. *"She does not like TV a lot, she likes play-ing. We play with a ball each other"* (mother of a girl aged three years and eight months).

*4.1.1.2. Parents' and principals' attitude.* Most mothers were satisfied with their children's diet and weight gain. They perceived that children were eating healthy food and fruits more than unhealthy snacks. *"Now I feel she is eating healthy food more than junk food. She eats a meal in the preschool and then eats with me at home and with her father in the afternoon. . ."* (mother of a girl aged five years and 10 months). Most mothers also reported that their children's weight is normal, and according to their doctor, their weight gain is satisfactory. *"His weight is often normal. . . the doctor is claiming he is on the growth line, above the growth line but not obesity, overall, his skeleton is huge"* (mother of a boy aged five years and one month). Parents and principals felt that children were eating foods better at preschool than at home. *"It seems she eats well here, but at home, her mother has to put food in her mouth"* (father of a girl aged four years and two months). Many parents were satisfied with their children's physical activity, and they said that their children were constantly jumping and their physical activity was at a high level. *"Her physical activity is tremendously high. Every moment she comes home, she plays a lot until she goes to bed"* (mother of a girl aged four years and nine months). Some mothers also complained about screen-based behaviors among their children. *"He watches ani-mations a lot. For example, when we go home from here, he watches animation until he sleeps"*

(mother of a boy aged five years and one month). Some mothers were inclined to have obese children, and they were not dissatisfied with their children's weight gain and eating behaviors. *"She is underweight and weak; she is slim . . ."* (mother of a 5.5-year-old girl).

*4.1.2. Knowledge. 4.1.2.1. Parents' knowledge.* Knowledge of the children's nutritional preferences and making healthy food were reported by most mothers. *". . . I even make pizza, not sausage. I use chicken and pizza cheese"* (mother of a girl aged five years and 10 months). *"He ate mixed meat, but now he does not eat that . . ."* (mother of a boy aged three years and six months).

*4.1.3. Beliefs. 4.1.3.1. Parents' beliefs.* Some parents believed that their children's weight would be normal. They believed that their children should stay fit and have a healthy weight. *"I always say a child has to be healthy, not overweight. . ."* (father of a 4-year-old girl).

*4.1.3.2. Principals' beliefs.* Principals believed that a child ate food well with other children and without coercion. *"If children don't eat at home due to coercion, they will have very well in the preschool. The most reason is that they eat meals together"* (principal, code 12).

**4.2. Enabling factors.** According to the data analysis, "parents', principals', and teachers' skills, rules and laws in the preschools, and also availability" were placed under the enabling factors.

*4.2.1. Skills. 4.2.1.1. Parents' and principals' skills.* Many parents reported performing healthy eating behaviors in front of their children. *"There is some food that her mother and I do not like, but we do not show that in the appearance . . . "*(father of a 4-year-old girl). Some parents have reduced their screen time. *"He used the phone maybe 1 or 2 hours a day, but now we have not used it for one year"* (mother of a 5-year-old boy). On the other hand, principals and one mother reported providing unhealthy snacks by mothers for their children. *". . . We had insisted on mothers to make natural juice. Some of them do it, and some of them don't . . ."* (principal, code 12). One of the skills of preschool principals and teachers was to pay attention to the health of preschool children *". . . cotlet (a mixture of meat, potatoes, and eggs), I don't really agree that a child eats it because of frying"* (principal, code 11).

*4.2.2. Laws and rules.* Preschool principals and some mothers stated that all children should consume preschool meals even very little. *". . . She/he says I don't eat food this morning, I say it is okay, eat two spoons (a little). Because it is a rule to eat it"* (principal, code 11). They also reported that there was a regular and similar diet plan in the preschool. *"The Preschool has a diet plan . . . based on the plan I provide the snack, and they eat it with each other"* (mother of a 4-year-old boy).

The principals mentioned that if the weather was favorable, they would play in the yard, and if the weather was rainy or snowy, they would play inside the preschool. *". . . If it is not snowy or rainy, they wear their clothes, and they play in the yard. If the weather is unfavorable. . ."* (principal, code 11). Principals also stated that according to a law in the preschool, there was a weekly food plan prepared by them and approved and signed by a nutritionist. However, they also reported that there was no nutritionist at the preschools to approve food plan; thus, they searched for this approval through mothers who knew an independent nutritionist or asked a nutritionist in the Welfare Organization. *"I write a weekly food plan and it is approved by a nutrition consultant. She has a seal"* (principal, code 11). There are some rules regarding the supervision of the Welfare Organization in preschools. The principal said that sometimes the inspector is referred from the Welfare Organization to monitor the preschool, and also some meetings and workshops are held by the Welfare Organization for the principals; however, attending meetings and workshops was not obligatory. *"The Welfare Organization Sometimes monitors the preschool. . . "*(principal, code 12).

*4.2.3. Availability.* Most mothers reported that screen appliances were available to children at home, and some children had a television in their room. *". . . He has a TV in his room and*

*lies down on his bed and watches an animated CD "* (mother of a boy aged five years and one month).

**4.3. Reinforcing factors.** According to the data analysis, "family support and effects, teachers' encouragement and influences, and peers' influences" were under the subcategory of feedback, encouragement, and influence of others and as reinforcing factors of weight management.

*4.3.1. Feedback, encouragement, and influence of others. 4.3.1.1. Family support and effects.* Most parents supported their children's physical activity and reported their children playing with their grandparents, uncle, peers, and brothers. *"Because she has an older brother and they run and play soccer a lot at home, and also we take them to the park most of the time"* (mother of a 5.5-year-old girl). On the other hand, parents' behaviors were encouraging to use screen by the children *". . . sometimes I said: play with my phone . . ."* (father of a 4-year-old girl). Parents' eating behaviors had an influence on the children's dietary behaviors, as many children refused or accepted some foods like their parents. *"If I say I don't eat something, she does not eat that too"* (mother of a girl aged three years and seven months).

*4.3.1.2. Teachers' encouragement and influences.* In preschool, children's behaviors could be influenced by their teachers, and some mothers reported affected children's eating and physical activity behaviors by their teachers. *"Teachers are also effective because my daughter constantly emphasizes what food items are useful and what food is not beneficial . . ."* (mother of a 4.5 year-old girl).

*4.3.1.3. Peers' influences.* Children were highly influenced by their peers regarding eating and physical activity behaviors as they followed their peers' behaviors and tried to behave like them. *"She said that one of my friends didn't eat food, I wanted not to eat it too"* (mother of a girl aged four years and nine months). Children were more encouraged to play and have physical activity when they had a playmate. *"Because my mother's next-door neighbor has a girl, she loves to go there and to play with her"* (mother of a girl aged four years and nine months).

## 5. Quality of child-parent relationship

**5.1. The effects of maternal mood status and quality of parental relationship on child's weight-related behaviors.** Some mothers reported affecting their children's eating and physical activity behaviors by their mood status and quality of the relationship between parents. They also stated that their children do not eat enough food or eat less affected by these factors. *"When I'm tired or upset, her behavior changes, her appetite will be less"* (mother of a girl aged four years and six months). Another mother said that she got divorced, which affected her son's eating: *"his father and I live separately . . . he ate little food at first, and he could not focus on everything . . ."* (mother a 5-year-old boy).

## Discussion

The aim of this study was to explore the perception and experiences of children's parents and principals regarding genetic, behavioral, environmental, and predisposing, reinforcing, and enabling factors on weight management in Iranian preschoolers. Based on our knowledge, these factors have not yet been explored from the point of view of Iranian children's parents and principals.

The first aim of the study was to investigate these factors from the parents' point of view. This study identified genetic, behavioral (e.g., food preferences, not eating some foods due to allergy or stomach reflux, physical activity and sedentary behaviors as children's actions) and (e.g., the effect of parents', peers', principals' and teachers' behavior and influence of grandparents' and neighbors' behaviors as actions of others) and environmental (e.g., home,

grandparents' home and preschool) factors from the epidemiological construct. Also, predisposing (e.g., child's attitude, parent's and principals' attitude, as well as parents' knowledge and parents' and principals' beliefs), enabling (e.g., parental skills and skills of the preschools' principals and teachers, rules and laws in the preschools, and availability), and reinforcing (e.g., family support and influences, teachers' encouragement and influences, and peers' influences) factors were identified from the educational and ecological construct. Some of these factors (e.g., genetic, food preferences, physical activity, sedentary behaviors, the effect of the behavior of parents, peers, teachers, and grandparents, parent's attitude and knowledge) have also been revealed in previous studies in children [24–35]. These similarities maybe indicate that these factors have a crucial effect on weight management promotion among preschoolers. Differences between some of our findings and other studies could be due to different families' cultures and the preschool's environment.

The second aim of the study was to investigate the factors from the point of view of preschool principals. Most of the identified factors by parents were also reported by principals; however, principals also stated factors that were not reported by the parents. Principals' additional factors were caring about children's health and diet in the preschool, existing preschool rules, and the influence of the weather on children's play. The combination of perspectives of parents and principals can be used to design an educational intervention for weight management promotion among preschoolers.

According to our findings, parents, especially mothers at home and principals at preschool, play a crucial role in promoting preschoolers' weight management. Consequently, future preschoolers' weight management promotion interventions should contain multiple components: one component focusing on mothers at home and one component focusing on principals at preschool. This is in line with studies reporting that multiple-component interventions are more effective than one-component [36, 37]. For instance, the home component can increase mothers' skills and knowledge and be a good role model for children. In contrast, the preschool component can focus on promoting principals' skills and teachers' role models.

In the current study, some mothers reported that their mood and the child-parent relationship quality could affect their children's eating and physical activity behaviors. A body of evidence supports this finding; there is an association between the quality of the parent-child relationship and obesogenic behaviors. Also, the emotional space created by parents may affect the association between parental practices and weight-related behaviors [38, 39].

## Strengths and limitations

The main strength of the present study is the fact that preschool principals were interviewed in addition to parents. Additionally, this study identified participants' attitude, beliefs, and behaviors that could be targeted in designing future interventions. One of the limitations of this study is that only two fathers were interviewed. Although mothers play an important role in shaping their children's behaviors in Iran, fathers also affect their behaviors in the home environment.

## Conclusion

Parents' and principals' experiences regarding preschoolers' weight management promotion confirm the genetic, behavioral, environmental, predisposing, enabling, and reinforcing factors of the PPM. Additionally, in the current study, "quality of child-parent relationship" was determined as a new factor affecting preschoolers' weight management promotion; however, it was not in the PPM. This finding may be related to culture and family relationship type among Iranian people and is suggested to be investigated in future studies.

## Supporting information

**S1 File. COREQ (COnsolidated criteria for REporting Qualitative research) checklist.**
(DOC)

## Acknowledgments

We are grateful to the participants in this study for sharing their time and their experiences to this research.

## Author Contributions

**Conceptualization:** Najmeh Hamzavi Zarghani, Fazlollah Ghofranipour, Eesa Mohammadi.

**Investigation:** Najmeh Hamzavi Zarghani.

**Methodology:** Najmeh Hamzavi Zarghani, Fazlollah Ghofranipour, Eesa Mohammadi.

**Project administration:** Fazlollah Ghofranipour.

**Validation:** Najmeh Hamzavi Zarghani, Eesa Mohammadi, Greet Cardon.

**Visualization:** Najmeh Hamzavi Zarghani.

**Writing – original draft:** Najmeh Hamzavi Zarghani.

**Writing – review & editing:** Najmeh Hamzavi Zarghani, Greet Cardon.

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
