## [Decision Letter · Decision Letter 0]

1 Oct 2021

PONE-D-20-18528Understanding the perceptions of parents and preschool principals on the determinants of weight management among Iranian preschoolers: A directed qualitative content analysisPLOS ONE

Dear Dr. Ghofranipour,

Thank you for submitting your manuscript to PLOS ONE. After careful consideration, we feel that it has merit but does not fully meet PLOS ONE’s publication criteria as it currently stands. Therefore, we invite you to submit a revised version of the manuscript that addresses the points raised during the review process.

The reviewers raised a number of concerns regarding the general organization and presentation of the manuscript, the inclusion of sufficient references in the background discussion, and several issues with the methodology and data analysis/presentation. Their comments can be viewed in full, below.

We look forward to receiving your revised manuscript.

Kind regards,

Natasha McDonald, PhD

Associate Editor

PLOS ONE

Journal Requirements:

Reviewers' comments:

Reviewer's Responses to Questions

**Comments to the Author**

1. Is the manuscript technically sound, and do the data support the conclusions?

Reviewer #1: Yes

Reviewer #2: Partly

2. Has the statistical analysis been performed appropriately and rigorously? 

Reviewer #1: N/A

Reviewer #2: Yes

3. Have the authors made all data underlying the findings in their manuscript fully available?

Reviewer #1: No

Reviewer #2: Yes

4. Is the manuscript presented in an intelligible fashion and written in standard English?

Reviewer #1: No

Reviewer #2: Yes

5. Review Comments to the Author

Reviewer #1: This manuscript provides an important contribution to the understanding of weight management among parents and school educators in an under-studied population (Iranian pre-school children). The use of the PRECEDE-PROCEED Model for framing the qualitative results is particularly helpful, although I have some queries about the choices made by the authors in their use of this model. I also have several other queries and comments about the paper, which I have outlined below.

General

1. There is inconsistent use of decimal places throughout the manuscript, i.e. in some places they are reported to two decimal places and one in others. It would be better if the authors picked one format and used it consistently.

2. The manuscript is too long. I have made some suggestions below as to where it could be shortened.

3. There are numerous grammatical errors in the manuscript. I have highlighted the major issues below, but the whole manuscript would benefit from English language revision.

Abstract

1. It would be helpful if the authors could briefly summarize the PROCEED-PRECEDE model in the background. I understand this is challenging because of word limits but given that it is central to the analysis of the data, it seems important that readers (who may only read the abstract) understand what it is.

Background

1. Second paragraph, first sentence – there would be a better flow if the authors followed the phrase “physical and psychological disorders” in their listing of the specific disorders themselves by first listing physical disorders, then listing psychological disorders. Mixing them together, as the authors have done, is disruptive.

2. Fourth paragraph, first sentence - needs grammatical revision for clarity.

3. Fourth paragraph, last sentence – reference 18 refers to Mexican American mothers, not Iranian parents. Please revise this sentence so that this is clear.

4. Sixth paragraph – the initialism PPM should be defined in full at its first mention.

5. Sixth paragraph – it is not clear why the authors chose to focus on the second and third phases of the PPM only. It would be helpful if they could provide some explanation for this decision – maybe this detail is more suited to the Methods section?

Methods – Study setting and participants

1. Can the authors please clarify what is meant by “The preschools were private, however, they were supervised by Tehran welfare organization”?

2. Table 1: The use of two decimal places here is unnecessary given the small numbers of participants in each of the groups. Please round up to one decimal place.

3. Table 1: It is unclear what the letters “A.s”, “B.s”, and “M.s” under the “Educational status of Participants” represent. Furthermore, is it necessary to break the participants’ educational status down to this level of detail for such a small number overall? These comments also apply to Table 2.

Methods – Data collection

1. Sentence three is very long and difficult to follow.

2. It is not necessary to provide the specific training and experience of the interviewer, suffice to say “trained researcher”.

3. Table 3: Question 1 (“How is your children’s eating status?”) appears to have translated poorly into English. Could the authors please clarify is this is a correct translation?

Methods – Data analysis

1. First sentence – repetition regarding participants consent for audio recording, please delete.

2. There is no mention of the phrase “direct content analysis” in this section, as is mentioned in the Abstract. This should be included so that the reader can be clear on the type of analysis used by the researchers.

Results

1. First paragraph, last sentence is an overstatement. The authors cannot factually state that the data were saturated, only that they considered it to be so. Please revise.

2. The results section is very long. Firstly, there is repetition regarding the PPM model itself littered throughout the entire results section. The manuscript could be shortened considerably if this were removed. Secondly, it may be beneficial for the authors to consider referring to Table 4 rather than repeating quotes in the table and text.

3. Section 4.1.1.2 – the points here regarding apartment living do not appear to fit with this code (Parents’ and principals’ attitude). They were already mentioned at section 2.2.4 but would also seem relevant under section 3.1.1. as they are related to the environment.

4. Section 4.2 (Reinforcing factors) – please rephrase “According to the participants…” at sentence one here. It was the researchers, not the participants, who were responsible for the analysis.

Discussion

1. This entire section needs substantial revision. There is frequent repetition of the results in the discussion, with little integrated discussion of findings of a whole. Rather than listing every finding in isolation, it would be helpful if the authors focused on what they consider to be the key results and how they may relate to each other.

Conclusion

1. The conclusion section should report the major takeaway findings of the study, not it’s novelty. It would be helpful if the authors outlined what specifically they consider to be the important sociocultural information provided by the study?

Reviewer #2: Authors have done a good job, but there are few things which need to be modified as suggested below. The authors should explain their choice of qualitative data collection, e.g why in-depth interviews and not focus group discussion and how they ended up with 15 participants. Its understandable that in qualitative we don't have sample size but at least it can be explained how the ended with only 15 as targeted (was there a saturation point?). I found the strengths and limitations sections difficult to understand, and the conclusion. These need to be thought through and improved

6. PLOS authors have the option to publish the peer review history of their article (what does this mean?). If published, this will include your full peer review and any attached files.

Reviewer #1: **Yes: **Éadaoin Máire Butler

Reviewer #2: No

---

## [Author Response · Author response to Decision Letter 0]

6 Feb 2022

Response to Reviewers 

We would like to thank our dear reviewers for their time and their valuable comments. We have seriously considered and addressed all reviewer comments giving point-to-point responses. All amendments have been highlighted in yellow in the text.

Comment:

1. There is inconsistent use of decimal places throughout the manuscript, i.e. in some places they are reported to two decimal places and one in others. It would be better if the authors picked one format and used it consistently.

Response: Yes, of course. The format was changed on page 3, line 55, and page 4 lines 77-8, page 7 line 146, and page 10 lines 183 as follows: 

The prevalence of overweight including obesity among preschoolers was found to be 32.0% and 10.0% to 20.6% in the United States and in Europe, respectively.

In a study conducted in Tehran, the capital of Iran, it was found that the prevalence of overweight and obesity among Tehranian children aged 3-6 years was 10.3% and 4.5% in girls and 9.8% and 4.7% in boys.

Variables Number Percent

Sex of participants 

Female 15 88.3

Male 2 11.7

Sex of child 

Female 11 64.7

Male 6 35.3

Educational status of Participants 

Bachelor and lower 13 76.5

Postgraduate education 4 23.5

Occupational status 

Housewife 4 23.5

Employed 13 76.5

Parents’ mean age was 35.1 (range: 28-42) years and principals’ mean age was 52.5 (range: 46-59) years.

Comment:

2. The manuscript is too long. I have made some suggestions below as to where it could be shortened.

Agreed. The results and discussion was revised to be shorter.

Comment:

3. There are numerous grammatical errors in the manuscript. I have highlighted the major issues below, but the whole manuscript would benefit from English language revision.

Agreed. The English language was revised by an editor. 

Comment:

Abstract

1. It would be helpful if the authors could briefly summarize the PROCEED-PRECEDE model in the background. I understand this is challenging because of word limits but given that it is central to the analysis of the data, it seems important that readers (who may only read the abstract) understand what it is.

Response: Yes, of course. The model was clarified on page 2, lines 28-33 as follows: 

The aim of the current study was to understand perceptions and experiences of Iranian parents and principals of preschool children on weight management based on the PRECEDE-PROCEED Model (PPM), a comprehensive structure for assessing health needs for designing, implementing, and evaluating health promotion and other public health programs. PRECEDE provides a structure for planning a targeted and focused public health program and PROCEED provides a structure for implementing and evaluating the program.

Comment:

Background

1. Second paragraph, first sentence – there would be a better flow if the authors followed the phrase “physical and psychological disorders” in their listing of the specific disorders themselves by first listing physical disorders, then listing psychological disorders. Mixing them together, as the authors have done, is disruptive.

Response: Agreed and revised on page 3, lines 60-2 as follows: 

The increased risk of physical disorders, such as cardiovascular diseases, cancer, and liver steatosis and also psychological disorders, like low self-esteem, body image concerns, depression, and weak socialization can be linked to overweight during childhood.

Comment:

2. Fourth paragraph, first sentence - needs grammatical revision for clarity.

Response: The correction was made on page 4, lines 74-5 as follows: 

With an 83 million population, Iran that is located in Western Asia is the 18th world’s most populous country. 

Comment:

3. Fourth paragraph, last sentence – reference 18 refers to Mexican American mothers, not Iranian parents. Please revise this sentence so that this is clear.

Response: The correction was made on page 4, lines 82-4 as follows: 

Therefore, understanding parents’ perceptions of their children’s weight status and its determinants are important to develop strategies and programs for weight management. 

Comment:

4. Sixth paragraph – the initialism PPM should be defined in full at its first mention.

Response: The correction was made on page 4, line 91 as follows: 

The PRECEDE-PROCEED Model (PPM) commonly provides practical guidance for the fields of health education and health promotion.

Comment:

5. Sixth paragraph – it is not clear why the authors chose to focus on the second and third phases of the PPM only. It would be helpful if they could provide some explanation for this decision – maybe this detail is more suited to the Methods section?

Response: Based on the reviewer’s comment, a clarification was added on page 4 and 5, lines 92-6 as follows: 

The current study mainly focused on the second (epidemiological assessment) and third (educational and ecological assessment) phases of the PPM, to better understand the health problem and potential modifiable strategies in the Iranian family and preschool context. Other stages are also important; however, are beyond the scope of the current study due to time and tools restrictions.

Comment:

Methods – Study setting and participants

1. Can the authors please clarify what is meant by “The preschools were private, however, they were supervised by Tehran welfare organization”?

Response: The clarification was made on page 6, lines 136-8 as follows: 

Although the preschools were supervised by Welfare Organization of Tehran, the costs of equipment and food preparation were provided by parents and preschool principals.

Comment:

2. Table 1: The use of two decimal places here is unnecessary given the small numbers of participants in each of the groups. Please round up to one decimal place.

Response: Agreed. The corrections were made on page 7 line 146 as follows: 

Variables Number Percent

Sex of participants 

Female 15 88.3

Male 2 11.7

Sex of child 

Female 11 64.7

Male 6 35.3

Educational status of Participants 

Bachelor and lower 13 76.5

Postgraduate education 4 23.5

Occupational status 

Housewife 4 23.5

Employed 13 76.5

Comment:

3. Table 1: It is unclear what the letters “A.s”, “B.s”, and “M.s” under the “Educational status of Participants” represent. Furthermore, is it necessary to break the participants’ educational status down to this level of detail for such a small number overall? These comments also apply to Table 2.

Response: The corrections were made on page 7 line 146 and 148 as follows: 

Variables Number Percent

Sex of participants 

Female 15 88.3

Male 2 11.7

Sex of child 

Female 11 64.7

Male 6 35.3

Educational status of Participants 

Bachelor and lower 13 76.5

Postgraduate education 4 23.5

Occupational status 

Housewife 4 23.5

Employed 13 76.5

Participants Age Sex Educational level Work experience

Principal 1 59 female Bachelor 35

Principal 2 46 female Bachelor 15

Comment:

Methods – Data collection

1. Sentence three is very long and difficult to follow.

Response: Agreed. The sentence was omitted and moved to Table 3. Page 8, lines 164.

Comment:

2. It is not necessary to provide the specific training and experience of the interviewer, suffice to say “trained researcher”.

Response: The correction was made on page 8, lines 156-7 as follows: 

The interviews were executed by the trained researcher in the preschool setting and lasted 15 to 35 minutes. 

Comment:

3. Table 3: Question 1 (“How is your children’s eating status?”) appears to have translated poorly into English. Could the authors please clarify is this is a correct translation?

Response: The clarification was made on page 8, line 164 as follows: 

1. Please explain your child’s nutrition status.

Comment:

Methods – Data analysis

1. First sentence – repetition regarding participants consent for audio recording, please delete.

Response: Based on the reviewer’s comment, on page 8, line 167-8 was revised as follows: 

Audio recording of the interviews was done by the first author and then, they were transcribed verbatim. 

Comment:

2. There is no mention of the phrase “direct content analysis” in this section, as is mentioned in the Abstract. This should be included so that the reader can be clear on the type of analysis used by the researchers.

Response: Agreed. The correction was made on page 9, lines 170-1 as follows: 

In line with the goals of the study, the researchers utilized the approach developed by Hsieh and Shannon [22] for directed content analysis.

Comment:

Results

1. First paragraph, last sentence is an overstatement. The authors cannot factually state that the data were saturated, only that they considered it to be so. Please revise.

Response: The suggested correction was made on page 10 and 11, lines 190-4 as follows: 

Data saturation was considered when the last couple of interviewees did not add new perceptions and sufficient data had been obtained regarding the object [23]. After 16 interviews, the codes were repeated and no new codes were found and the data were considered to be saturated and data collection was stopped. 

Comment:

2. The results section is very long. Firstly, there is repetition regarding the PPM model itself littered throughout the entire results section. The manuscript could be shortened considerably if this were removed. Secondly, it may be beneficial for the authors to consider referring to Table 4 rather than repeating quotes in the table and text.

Response: As suggested, the results section was shortened and repetitions were omitted. 

Comment:

3. Section 4.1.1.2 – the points here regarding apartment living do not appear to fit with this code (Parents’ and principals’ attitude). They were already mentioned at section 2.2.4 but would also seem relevant under section 3.1.1. as they are related to the environment.

Response: Thank you for your comment. We agree that apartment living do not fit with Parents’ and principals’ attitude, however we believe it is related to “Action of neighbors”. 

Comment:

4. Section 4.2 (Reinforcing factors) – please rephrase “According to the participants…” at sentence one here. It was the researchers, not the participants, who were responsible for the analysis. 

Response: Agreed and revised on page 18 lines 361-3 and page 20, lines 400-3 as follows: 

4.2. Enabling factors

According to the data analysis, "parents’, principals’, and teachers’ skills, rules and laws in the preschools, and also availability" were placed under the enabling factors. 

4.3. Reinforcing factors

According to the data analysis, “family support and effects, teachers’ encouragement and influences, and peers’ influences” were under the subcategory of feedback, encouragement, and influence of others and as reinforcing factors of weight management.

Comment:

Discussion

1. This entire section needs substantial revision. There is frequent repetition of the results in the discussion, with little integrated discussion of findings of a whole. Rather than listing every finding in isolation, it would be helpful if the authors focused on what they consider to be the key results and how they may relate to each other.

Response: The discussion was revised accordingly on page 22-24 lines 437-480.

Discussion 

The aim of this study was to explore the perception and experiences of children’s parents and principals regarding genetic, behavioral, environmental, and also predisposing, reinforcing and enabling factors on weight management in Iranian preschoolers. Based on our knowledge, these factors have not yet been explored from the point of view of Iranian children’s parents and principals. 

The first aim of the study was to investigate these factors from the point of view of parents. This study identified genetic, behavioral (e.g., food preferences, not eating some foods due to allergy or stomach reflux and also physical activity and sedentary behaviors as children’s actions) and (e.g., the effect of parents’, peers’, principals’ and teachers’ behavior and also influence of grandparents’ and neighbors’ behaviors as actions of others) and environmental (e.g., home, grandparents’ home and preschool) factors from the epidemiological construct. Also, predisposing (e.g., child’s attitude, parent’s and principals’ attitude, as well as parents’ knowledge and parents’ and principals’ beliefs), enabling (e.g., parental skills and skills of the principals and teachers, rules and laws in the preschools, and availability), and reinforcing (e.g., family support and influences, teachers’ encouragement and influences, and peers’ influences) factors were identified from the educational and ecological construct. Some of these factors (e.g., genetic, food preferences, physical activity, sedentary behaviors, the effect of the behavior of parents, peers, teachers, and grandparents, parent’s attitude and knowledge) have also been revealed in previous studies in children [24-35]. These similarities maybe indicate that these factors have a crucial effect on weight management promotion among preschoolers. Differences between some of our findings and other studies could be due to different cultures of families and the preschool’s environment. 

The second aim of the study was to investigate the factors from the point of view of preschool principals. Most of the identified factors by parents were also reported by principals; however, principals also stated factors that were not reported by the parents. Additional factors reported by principals were caring about children’s health and diet in the preschool, existing rules in preschools, and the influence of the weather on children’s play. The combination of perspectives of parents and principals can be used to design an educational intervention for weight management promotion among preschoolers. 

In general, according to our findings, parents, especially mothers, at home environment and principals at preschool play a crucial role to promote preschoolers’ weight management. Consequently, future interventions for preschoolers’ weight management promotion should contain multiple components: one component focusing on mothers at home and one component focusing on principals at preschool. This is in line with studies reporting that multiple-component interventions are more effective than one-component interventions [36, 37]. For instance, the home component can be considered to increase mothers’ skills and knowledge and be a good role model for children, whereas the preschool component can focus on the promotion of principals’ skills and teachers’ role models. 

In the current study, some mothers reported that their mood and also the parents-children relationship quality can affect their children’s eating and physical activity behaviors. A body of evidence supports this finding; there is an association between the quality of the parent-child relationship and obesogenic behaviors. Also, the emotional space created by parents may affect the association between parental practices and weight-related behaviors [38, 39]. 

Comment:

Conclusion

1. The conclusion section should report the major takeaway findings of the study, not it’s novelty. It would be helpful if the authors outlined what specifically they consider to be the important sociocultural information provided by the study?

Response: Agreed and revised on page 24, lines 488-494 and also page 2 and 3 lines 46-51 as follows: 

Conclusion

Parents’ and principals’ experiences regarding preschoolers’ weight management promotion confirm the genetic, behavioral, environmental, predisposing, enabling, and reinforcing factors of the PPM. Additionally, in the current study, “quality of child-parent relationship” was determined as a new factor affecting preschoolers’ weight management promotion; however, it was not in the PPM. This finding may be related to culture and family relationship type among Iranian people and is suggested to be investigated in future studies.

Comment:

Reviewer #2: Authors have done a good job, but there are few things which need to be modified as suggested below. The authors should explain their choice of qualitative data collection, e.g why in-depth interviews and not focus group discussion and how they ended up with 15 participants. Its understandable that in qualitative we don't have sample size but at least it can be explained how the ended with only 15 as targeted (was there a saturation point?). I found the strengths and limitations sections difficult to understand, and the conclusion. These need to be thought through and improved

Response: Thanks for your comment. 

Firstly, in qualitative studies data collection method is not predetermined and fixed and is influenced by such factors as objects of the study, participant’s willingness and their cultural background. In the current study, participants were inclined to individual interviews. In addition, Iranian mothers culturally tend to be reluctant to discuss their children's behavior in public. Secondly, depth interviews were chosen above other methods to get enough detail. 

Response: The suggested correction was made on page 10 and 11, lines 190-4 as follows:

Data saturation was considered when the last couple of interviewees did not add new perceptions and sufficient data had been obtained regarding the object [23]. After 16 interviews, the codes were repeated and no new codes were found and the data were considered to be saturated and data collection was stopped. 

Response: Agreed and revised on page 264 lines 481-494 and also page 2 and 3 lines 46-51 as follows: 

Strengths and limitations 

The main strength of the present study is the fact that preschool principals were interviewed in addition to parents. Additionally, this study identified participants’ attitude, believes, and behaviors that could be targeted in designing future interventions. One of the limitations of this study is that only two fathers were interviewed. Although in Iran, mothers play an important role in shaping the behaviors of their children, fathers also affect their behaviors in the home environment.

Conclusion

Parents’ and principals’ experiences regarding preschoolers’ weight management promotion confirm the genetic, behavioral, environmental, predisposing, enabling, and reinforcing factors of the PPM. Additionally, in the current study, “quality of child-parent relationship” was determined as a new factor affecting preschoolers’ weight management promotion; however, it was not in the PPM. This finding may be related to culture and family relationship type among Iranian people and is suggested to be investigated in future studies.

---

## [Decision Letter · Decision Letter 1]

25 Mar 2022

PONE-D-20-18528R1Understanding the perceptions of parents and preschool principals on the determinants of weight management among Iranian preschoolers: A directed qualitative content analysisPLOS ONE

Dear Dr. Ghofranipour,

Thank you for submitting your manuscript to PLOS ONE. After careful consideration, we feel that it has merit but does not fully meet PLOS ONE’s publication criteria as it currently stands. Therefore, we invite you to submit a revised version of the manuscript that addresses the points raised during the review process.

For transparency purposes, I wish to alert you that I previously acted as reviewer of this manuscript. As the previous second reviewer was unable to complete a review of your revised manuscript within the required timeframe, a third reviewer was enlisted to consider your revised manuscript against the PLOS ONE publication criteria. This reviewer raised two major concerns, one of which relates to the English language standard of the manuscript. Although I am aware that you have already had the manuscript assessed by an English language editor, I agree with the reviewer that there are numerous remaining grammatical issues. Of particular concern, some of these grammatical issues diminish interpretation of the content. In order for this manuscript to meet PLOS ONE's publication criteria, there will need to be substantial improvements to the English language standard. On that basis, I would suggest that you have your revised manuscript assessed by another English language editor. In addition, the reviewer has provided some helpful suggestions regarding grammatical improvements to the manuscript that you may wish to incorporate into your revision. 

We look forward to receiving your revised manuscript.

Kind regards,

Dr Éadaoin Butler

Academic Editor

PLOS ONE

Reviewers' comments:

Reviewer's Responses to Questions

**Comments to the Author**

1. If the authors have adequately addressed your comments raised in a previous round of review and you feel that this manuscript is now acceptable for publication, you may indicate that here to bypass the “Comments to the Author” section, enter your conflict of interest statement in the “Confidential to Editor” section, and submit your "Accept" recommendation.

Reviewer #3: All comments have been addressed

2. Is the manuscript technically sound, and do the data support the conclusions?

Reviewer #3: Yes

3. Has the statistical analysis been performed appropriately and rigorously? 

Reviewer #3: Yes

4. Have the authors made all data underlying the findings in their manuscript fully available?

Reviewer #3: Yes

5. Is the manuscript presented in an intelligible fashion and written in standard English?

Reviewer #3: No

6. Review Comments to the Author

Reviewer #3: The paper is framed using the PPM and includes important results and implications for the target audience. However, the manuscript should be organized to provide an overview of findings with a PPM figure. Also, the paper must be carefully edited for grammatical errors.

7. PLOS authors have the option to publish the peer review history of their article (what does this mean?). If published, this will include your full peer review and any attached files.

Reviewer #3: No

---

## [Author Response · Author response to Decision Letter 1]

15 Apr 2022

We would like to thank the reviewer for her/him time and valuable comments. We have seriously considered and addressed all the reviewer’s comments giving point-to-point responses. All amendments have been highlighted in yellow in the text.

Comment:

Major issue

1. A major issue is that the paper still includes many grammatical errors and needs to be revised by an editor to improve clarity.

Response: Agreed. The English language was revised by an editor. 

2. Also, I am unclear about the use of the same questions for parents/principals- wouldn’t the questions need to different for principals as educators not parents?

Response: Indeed questions differed between the groups of respondents. This was clarified by adding an audience column on page 8, line 166 as follows:

Table 3: Interview Questions

Constructs of PPM Interview Questions Audiences

Behavioral factors 1. Please explain your child’s nutrition status. parents

 2. Please explain your child/ children’s physical activity on weekdays/ weekends. parents/principals

 3. How do you manage your child/ children’s weight status? parents/principals

Predisposing factors 4. How do you assess your children’s weight? (Based on the practitioner’s assessment or your own assessment) parents

 5. Why do you manage your child/ children’s weight status? parents/principals

 6. What is your belief regarding the weight status of children aged 3-5 years? parents/principals

Enabling factors 7. Please tell me about the rules in your home to manage your child’s weight. parents

 8. Please tell me about diet and physical activity plans in the preschool. parents/principals

 9. What are the rules in the preschool to improve children’s weight management? principals

 10. What problems do you encounter to manage your child’s weight? parents

Reinforcement factors 11. Please tell me how you influence the dietary intake and physical activity of your children. parents

 12. Please tell me how teachers and peers influence your child/ children’s eating habits and physical activity. parents/principals

Minor issues 

Comment:

Abstract

1. Results- reword “Obese children” to children with obesity. 

 Response: Agreed. Based on your next comment, the sentences were revised completely. 

2. Results may be better understood if described within the PRECEDE sections.

Response: Agreed and revised on page 2 lines 36-46 as follows:

This study identified genetic, behavioral (e.g., food preferences, physical activity, sedentary behaviors, the effect of parents’, peers’, principals’ and teachers’ behavior and influence of grandparents’ and neighbors’ behaviors) and environmental (e.g., home, grandparents’ home and preschool) factors from the epidemiological construct. Additionally, predisposing (e.g., child’s attitude, parent’s and principals’ attitude, as well as parents’ knowledge and parents’ and principals’ beliefs), enabling (e.g., parental skills and skills of the principals and teachers, rules and laws in the preschools, and availability), and reinforcing (e.g., family support and influences, teachers’ encouragement and influences, and peers’ influences) factors were identified from the educational and ecological construct. 

Background

1. Lines 106, 109 – Sentences needs citations 

 Response: Of course. The citations were added on page 5, lines 106-109.

Study design

1. Need to cite content analysis approach (line 125) 

 Response: Of course. The citations was added on page 6, lines 125.

2. Line 127 We used individual semi-structured interviews with open-ended questions, and assessed play equipment and documents, such as the diet and physical activity plans in preschools. This triangulation method increases the credibility and conformability of the study and the understanding of various aspects of weight management promotion 

Response: The correction was made on page 6, lines 126-130 as follows: 

 We used individual semi-structured interviews with open-ended questions, and assessed play equipment and documents, such as the diet and physical activity plans in preschools. This triangulation method increases the credibility and conformability of the study and the understanding of various aspects of weight management promotion.

3. Line 136 Preschools in six areas of Tehran

4. line 137 supervised by the Welf

5. Line 142 remove the following it is unclear “and if needed, the fathers were also to continue interview.”

6. Line 143 add being a principal as criterion 

7. Line 153 change to Interview questions

Response: Agreed. All of mentioned errors were corrected on pages 6, 7 and 8, lines 135, 137, 141, 142, and 155.

8. Line 158 After the participants’ answered, the researcher utilized probe questions to explore participants’ experiences of the concepts by asking questions, such as “What do you mean?, Please explain more about …?, Would you give me an example to understand what you mean?”. After transcription and review of some interviews, to clarify ambiguities, the researcher called the participants and asked them for more details.

 Response: Agreed and revised on page 8, lines 159-164 as follows: 

 After the participants answered, the researcher utilized probe questions to explore participants’ experiences of the concepts by asking some questions, such as “What do you mean?, Please explain more about …?, Could you give me an example to understand what you mean?”. After transcription and review of some interviews, to clarify ambiguities, the researcher called the participants and asked them for more details.

9. Line 164 Table 3. Interview Questions – how did you ask the principals the questions were they revised to fit their job vs. their child?

 Response: We didn’t ask the principals regarding fitting principals’ job vs. their child since there is not any questions to fit principals’ job vs. their child in PRECEDE-PROCEED model. We just asked them “Please tell me how teachers and peers influence your child/ children’s eating habits and physical activity”. 

10. Be consistent with wording- first author or main researcher?

Response: Agreed. First author and main researcher were replaced by the researcher in the manuscript on page 6, 8 and 9, lines 120, 160,163, 168, and 172.

Results

1. This section could benefit from a figure to show findings based on PPM

Response: as suggested a figure was designed and added on page 22, line 439 as follow: 

 Figure 1: PRECEDE-PROCEED model to understand perceptions and experiences of Iranian parents and principals of preschool children on weight management 

2. Line 221 if weather was good 

3. Line 237 Parents also reported that was they were responsible to take their children to the preschool by car or on foot.

4. Line 426 Quality of child-parents relationship

Response: Agreed. All of mentioned comments were corrected on pages 12 and 13, 21 lines 223, 239, and 429.

---

## [Decision Letter · Decision Letter 2]

31 May 2022

PONE-D-20-18528R2Understanding the perceptions of parents and preschool principals on the determinants of weight management among Iranian preschoolers: A directed qualitative content analysisPLOS ONE

Dear Dr. Ghofranipour,

Thank you for submitting your manuscript to PLOS ONE. After careful consideration, we feel that it has merit but does not fully meet PLOS ONE’s publication criteria as it currently stands. Therefore, we invite you to submit a revised version of the manuscript that addresses the points raised during the review process.

The remaining changes I would like made are solely English language edits. Once these changes have been addressed, the paper can be accepted for publication. These edits are:

**Introduction**

"With a **population**
**of** 83 million **people**, Iran**,**
that is located in Western Asia, is the 18th world’s **18^th^** most populous country.”

**Study setting and participants**

“…The inclusion criterion was being a principal **or** parent of preschoolers aged 3-5 years.” 

(Note: This change does not apply if the criterion did indeed specify that they had to be **both** a principal **and** a parent, although I suspect this is not the case.)

**Results**

**3.1.3. Preschool**

Children spend most of the days in a week and most of the hours in a day in the preschool **and** were eating breakfast, lunch, and snacks there.

**4.1.1.2. Parents’ and principals’ attitude**

Some mothers **tended** that their children were obese, and they were not dissatisfied with  their children’s weight gain and eating behaviors. "She is underweight and weak; she is slim …"

(Note: Please use a different phrase here as "tended" does not make any sense in this context.)

We look forward to receiving your revised manuscript.

Kind regards,

Éadaoin Butler

Guest Editor

PLOS ONE

Journal Requirements:

Reviewers' comments:

Reviewer's Responses to Questions

**Comments to the Author**

1. If the authors have adequately addressed your comments raised in a previous round of review and you feel that this manuscript is now acceptable for publication, you may indicate that here to bypass the “Comments to the Author” section, enter your conflict of interest statement in the “Confidential to Editor” section, and submit your "Accept" recommendation.

Reviewer #2: All comments have been addressed

2. Is the manuscript technically sound, and do the data support the conclusions?

Reviewer #2: Yes

3. Has the statistical analysis been performed appropriately and rigorously? 

Reviewer #2: Yes

4. Have the authors made all data underlying the findings in their manuscript fully available?

Reviewer #2: Yes

5. Is the manuscript presented in an intelligible fashion and written in standard English?

Reviewer #2: Yes

6. Review Comments to the Author

Reviewer #2: The author have addressed all comments as required. Statistical section is well structured, data are well presented and results are clear. The language is well understood

7. PLOS authors have the option to publish the peer review history of their article (what does this mean?). If published, this will include your full peer review and any attached files.

Reviewer #2: **Yes: **Mary Vincent Mosha

---

## [Author Response · Author response to Decision Letter 2]

4 Jun 2022

Introduction

Comment:

"With a population of 83 million people, Iran, that is located in Western Asia, is the 18th world’s 18th most populous country.”

Response: The correction was made on page 4, lines 74-75 accordingly:

With a population of 83 million people, Iran, that is located in Western Asia, is the world’s 18th most populous country.

Study setting and participants

Comment:

“…The inclusion criterion was being a principal or parent of preschoolers aged 3-5 years.”

(Note: This change does not apply if the criterion did indeed specify that they had to be both a principal and a parent, although I suspect this is not the case.)

Response: Agreed. The correction was made on page 7, lines 142 accordingly:

The inclusion criterion was being a principal or parent of preschoolers aged 3-5 years.

Results

3.1.3. Preschool

Comment:

Children spend most of the days in a week and most of the hours in a day in the preschool and were eating breakfast, lunch, and snacks there.

Response: The correction was made on page 16, lines 309 accordingly:

Children spend most of the days in a week and most of the hours in a day in the preschool and were eating breakfast, lunch, and snacks there.

4.1.1.2. Parents’ and principals’ attitude

Comment:

Some mothers tended that their children were obese, and they were not dissatisfied with their children’s weight gain and eating behaviors. "She is underweight and weak; she is slim …"

(Note: Please use a different phrase here as "tended" does not make any sense in this context.)

Response: Agreed. The correction was made on page 18, lines 346 as follow:

Some mothers were inclined to have obese children, and they were not dissatisfied with their children’s weight gain and eating behaviors.

---

## [Editor Report · Decision Letter 3]

8 Jun 2022

Understanding the perceptions of parents and preschool principals on the determinants of weight management among Iranian preschoolers: A directed qualitative content analysis

PONE-D-20-18528R3

Dear Dr. Ghofranipour,

We’re pleased to inform you that your manuscript has been judged scientifically suitable for publication and will be formally accepted for publication once it meets all outstanding technical requirements.

Kind regards,

Éadaoin Butler

Guest Editor

PLOS ONE
---

## [Editor Report · Acceptance letter]

13 Jun 2022

PONE-D-20-18528R3 

Understanding the perceptions of parents and preschool principals on the determinants of weight management among Iranian preschoolers: A directed qualitative content analysis

Dear Dr. Ghofranipour:

I'm pleased to inform you that your manuscript has been deemed suitable for publication in PLOS ONE. Congratulations! Your manuscript is now with our production department. 

Kind regards, 

on behalf of

Dr. Éadaoin Butler 

Guest Editor

PLOS ONE